# SMALL STEPS AND GIANT LEAPS:
# MINIMAL NEWTON SOLVERS FOR DEEP LEARNING

## ABSTRACT

We propose a fast second-order method that can be used as a drop-in replacement for current deep learning solvers. Compared to stochastic gradient descent (SGD), it only requires two additional forward-mode automatic differentiation operations per iteration, which has a computational cost comparable to two standard forward passes and is easy to implement. Our method addresses long-standing issues with current second-order solvers, which invert an approximate Hessian matrix every iteration exactly or by conjugate-gradient methods, procedures that are much slower than a SGD step. Instead, we propose to keep a single estimate of the gradient projected by the inverse Hessian matrix, and update it once per iteration with just two passes over the network. This estimate has the same size and is similar to the momentum variable that is commonly used in SGD. No estimate of the Hessian is maintained. We first validate our method, called CURVEBALL, on small problems with known solutions (noisy Rosenbrock function and degenerate 2-layer linear networks), where current deep learning solvers struggle. We then train several large models on CIFAR and ImageNet, including ResNet and VGG-f networks, where we demonstrate faster convergence with no hyperparameter tuning. We also show our optimiser's generality by testing on a large set of randomly-generated architectures.

## 1 INTRODUCTION

Stochastic Gradient Descent (SGD) and back-propagation (LeCun et al., 1998) are the algorithmic backbone of current deep network training. The success of deep learning demonstrates the power of this combination, which has been successfully applied on various tasks with large datasets and very deep networks (He et al., 2016).

Yet, while SGD has many advantages, speed of convergence (in terms of number of iterations) is not necessarily one of them. While individual SGD iterations are very quick to compute and lead to rapid progress at the beginning of the optimisation, it soon reaches a slower phase where further improvements are achieved slowly. This can be attributed to entering regions of the parameter space where the objective function is poorly scaled. In such cases, rapid progress would require vastly different step sizes for different directions in parameter space, which SGD cannot deliver.

Second-order methods, such as Newton's method and its variants, eliminate this issue by rescaling the gradient according to the local curvature of the objective function. For a scalar loss in $\mathbb{R}$, this rescaling takes the form $H^{-1}J$ where $H$ is the Hessian matrix (second-order derivatives) or an approximation of the local curvature in the objective space, and $J$ is the gradient of the objective. They can in fact achieve local scale-invariance (Wright & Nocedal, 1999, p. 27), and make provably better progress in the regions where gradient descent stalls. While they are unmatched in other domains, there are several obstacles to their application to deep models. First, it is impractical to invert or even store the Hessian, since it grows quadratically with the number of parameters, and there are typically millions of them. Second, any Hessian estimate is necessarily noisy and ill-conditioned due to stochastic sampling, to which classic inversion methods such as conjugate-gradient are not robust.

In this paper, we propose a new algorithm that can overcome these difficulties and make second order optimisation practical for deep learning. We show in particular how to avoid the storage of any estimate of the Hessian matrix or its inverse. Instead, we treat the computation of the Newton update, $H^{-1}J$, as solving a linear system that itself can be solved via gradient descent. The cost of solving this system is amortized over time by interleaving its steps with the parameter update steps. Our

proposed method adds little overhead, since a Hessian-vector product can be implemented for modern networks with just two steps of automatic differentiation. Interestingly, we show that our method is equivalent to momentum SGD (also known as the heavy-ball method) with a single additional term, accounting for curvature. For this reason we named our method CURVEBALL. Unlike other proposals, the total memory footprint is as small as that of momentum SGD.

This paper is structured as follows. We introduce relevant technical background in sec. 2, and present our method in sec. 3. We evaluate our method and show experimental results in sec. 4. Related work is discussed in sec. 5. Finally we summarise our findings in sec. 6.

## 2 BACKGROUND

In order to make the description of our method self-contained, we succinctly summarise a few standard concepts in optimisation. Our goal is to find the optimal parameters of a model (*e.g.* a neural network) $\phi : \mathbb{R}^p \rightarrow \mathbb{R}^o$, with $p$ parameters $w \in \mathbb{R}^p$ and $o$ outputs (the notation does not show the dependency on the training data, which is subsumed in $\phi$ for compactness). The quality of the outputs is evaluated by a loss function $L : \mathbb{R}^o \rightarrow \mathbb{R}$, so finding $w$ is reduced to the optimisation problem: [1]

$$w^* = \arg\min_w L(\phi(w)) = \arg\min_w f(w). \tag{1}$$

Perhaps the simplest algorithm to find an optimum (or at least a stationary point) of eq. 1 is gradient descent (GD). GD updates the parameters using the iteration $w \leftarrow w - \beta J(w)$, where $\beta > 0$ is the learning rate and $J(w) \in \mathbb{R}^p$ is the gradient (or Jacobian) of the objective function $f$ with respect to the parameters $w$. A useful variant is to augment GD with a *momentum* variable $z$ (Polyak, 1964), which can be interpreted as a decaying average of past gradients:[2]

$$z \quad \leftarrow \quad \rho z - \beta J(w) \tag{2}$$
$$w \quad \leftarrow \quad w + z \tag{3}$$

with a momentum parameter $\rho$. Momentum GD, as given by eq. 2-3, can be shown to have faster convergence than GD for convex functions, remaining stable under higher learning rates, and exhibits somewhat better resistance to poor scaling of the objective function (Nesterov, 2013; Goh, 2017). One important aspect is that these advantages cost almost no additional computation and only a modest additional memory, which explains why it is widely used in practice.

In neural networks, GD is usually replaced by its stochastic version (SGD), where at each iteration one computes the gradient not of the model $f = L(\phi(w))$, but of the model $f_t = L_t(\phi_t(w))$ assessed on a small batch of samples, drawn at random from the training set.

### 2.1 SECOND-ORDER OPTIMISATION

As mentioned in section 1, the Newton method is similar to GD, but steers the gradient by the inverse Hessian matrix, computing $H^{-1}J$ as a descent direction. However, inverting the Hessian may be numerically unstable or the inverse may not even exist. To address this issue, the Hessian is usually regularized with a parameter $\lambda > 0$, obtaining what is known as the *Levenberg* (Moré, 1978) method:

$$z \quad = \quad -(H + \lambda I)^{-1}J, \tag{4}$$
$$w \quad \leftarrow \quad w + z, \tag{5}$$

where $H \in \mathbb{R}^{p \times p}$, $J \in \mathbb{R}^p$ and $I \in \mathbb{R}^{p \times p}$ is the identity matrix. Note that, unlike for momentum GD (eq. 2), the new step $z$ is independent of the previous step. To avoid burdensome notation, we omit the $w$ argument in $H(w)$ and $J(w)$, but they must be recomputed at each iteration. Intuitively, the effect of eq. 4 is to rescale the step appropriately for different directions — directions with high curvature require small steps, while directions with low curvature require large steps to make progress.

Note also that Levenberg's regularization loses the scale-invariance of the original Newton method, meaning that rescaling the function $f$ changes the scale of the gradient and hence the regularised descent direction chosen by the method. An alternative that alleviates this issue is *Levenberg-Marquardt*, which replaces $I$ in eq. 4 with $\text{diag}(H)$. For non-convex functions such as deep networks, these methods only converge to a local minimum when the Hessian is positive-semidefinite (PSD).

---

[1]We omit the optional regulariser term for brevity, but this does not materially change our derivations.

[2]Some sources use an alternative form with $\beta$ in eq. 3 instead (equivalent by a re-parametrisation of $z$),

## 2.2 AUTOMATIC DIFFERENTIATION AND BACK-PROPAGATION

In order to introduce fast computations involving the Hessian, we must take a short digression into how Jacobians are computed. The Jacobian of $L(\phi(w))$ (eq. 1) is generally computed as $J = J_\phi J_L$ where $J_\phi \in \mathbb{R}^{p \times o}$ and $J_L \in \mathbb{R}^{o \times 1}$ are the Jacobians of the model and loss, respectively. In practice, a Jacobian is never formed explicitly, but Jacobian-vector products $Jv$ are implemented with the back-propagation algorithm. We define

$$\overleftarrow{\text{AD}}(v) = Jv \tag{6}$$

as the *reverse-mode automatic differentiation* (RMAD) operation, commonly known as *back-propagation*. Note that, because the loss is a scalar function, the starting projection vector $v$ typically used in gradient descent is a scalar and we set $v = 1$. For intermediate computations, however, it is generally a (vectorized) tensor of gradients (see eq. 9).

A perhaps lesser known alternative is *forward-mode automatic differentiation* (FMAD), which computes a vector-Jacobian product, from the other direction:

$$\overrightarrow{\text{AD}}(v') = v'J \tag{7}$$

This variant is less commonly-known in deep learning as RMAD is appropriate to compute the derivatives of a scalar-valued function, such as the learning objective, whereas FMAD is more appropriate for vector-valued functions of a scalar argument. However, we will show later that FMAD is relevant in calculations involving the Hessian.

The only difference between RMAD and FMAD is the direction of associativity of the multiplication: FMAD propagates gradients in the forward direction, while RMAD (or back-propagation) does it in the backward direction. For example, for the composition of functions $a \circ b \circ c$,

$$\overrightarrow{\text{AD}}_{a \circ b \circ c}(v) = ((vJ_a) J_b) J_c \tag{8}$$

$$\overleftarrow{\text{AD}}_{a \circ b \circ c}(v') = J_a (J_b (J_c v')) \tag{9}$$

Because of this, both operations have similar computational overhead, and can be implemented similarly.

## 2.3 FAST HESSIAN-VECTOR PRODUCTS

Since the Hessian of learning objectives involving deep networks is not necessarily positive semi-definite (PSD), it is common to use a surrogate matrix with this property, which prevents second-order methods from being attracted to saddle-points (this problem is discussed by Dauphin et al. (2014)). One of the most widely used is the Gauss-Newton approximation (Martens & Grosse, 2015; Botev et al., 2017; Wright & Nocedal, 1999, p. 254):

$$\hat{H} = J_\phi H_L J_\phi^T, \tag{10}$$

When $H_L$ is PSD, which is the case for all convex losses (*e.g.* logistic loss, $L^p$ distance), the resulting $\hat{H}$ is PSD by construction. For the method that we propose, and indeed for any method that implicitly inverts the Hessian (or its approximation), only computing Hessian-vector products $\hat{H}v$ is required. As such, eq. 10 takes a very convenient form:

$$\hat{H}v = J_\phi \left( H_L \left( J_\phi^T v \right) \right) \tag{11}$$

$$= \overleftarrow{\text{AD}}_\phi \left( H_L \left( \overrightarrow{\text{AD}}_\phi(v) \right) \right). \tag{12}$$

The cost of eq. 12 is thus equivalent to that of two back-propagation operations. This is similar to a classic result (Pearlmutter, 1994; Schraudolph, 2002), but written in terms of common automatic differentiation operations. The intermediate matrix-vector product $H_L u$ has negligible cost: for example, for the squared-distance loss, $H_L = 2I \Rightarrow H_L u = 2u$. Similarly, for the multinomial logistic loss we have $H_L = \text{diag}(p) - pp^T \Rightarrow H_L u = p \odot u - p(p^T u)$, where $p$ is the vector of predictions from a softmax layer and $\odot$ is the element-wise product. These products thus require only element-wise operations.

| Algorithm 1: CURVEBALL (proposed). | Algorithm 2: Simplified Hessian-free method. |
|---|---|
| 1: $z_0 = \mathbf{0}$ | 1: **for** $t = 0, ..., T - 1$ **do** |
| 2: **for** $t = 0, ..., T - 1$ **do** | 2: $\quad z_0 = -J(w_t)$ |
| 3: $\quad \Delta_z = \hat{H}(w_t)z_t + J(w_t)$ | 3: $\quad$ **for** $r = 0, ..., R - 1$ (or convergence) **do** |
| 4: $\quad z_{t+1} = \rho z_t - \beta \Delta_z$ | 4: $\quad\quad z_{r+1} = \text{CG}(z_r, \hat{H}(w_t)z_r, J(w_t))$ |
| 5: $\quad w_{t+1} = w_t + z_{t+1}$ | 5: $\quad$ **end for** |
| 6: **end for** | 6: $\quad w_{t+1} = w_t + z_R$ |
|  | 7: **end for** |

## 3 METHOD

This section presents our main contribution: a method that minimizes a second-order Taylor expansion of the objective (like the Newton variants from section 2.1), but at a much reduced computational and memory cost, suitable for very large-scale problems. The result of taking a step $z$ away from a starting point $w$ can be modelled with a second-order Taylor expansion of the objective $f$:

$$f(w + z) \simeq \hat{f}(w, z) = f(w) + z^T J(w) + \tfrac{1}{2} z^T H(w) z \tag{13}$$

Most second-order methods seek the update $z$ that minimizes $\hat{f}$, by ignoring the higher-order terms:

$$z = \arg \min_{z'} \hat{f}(z') = \arg \min_{z'} \tfrac{1}{2} z'^T \hat{H} z' + z'^T J \tag{14}$$

In general, a step $z$ is found by minimizing eq. 14, either via explicit inversion $\hat{H}^{-1}J$ (Martens & Grosse, 2015; Botev et al., 2017) or the conjugate-gradient (CG) method (Martens, 2010). The later approach, called the Hessian-free method (also Truncated Newton or Newton-CG (Wright & Nocedal, 1999, p. 168)) is the most economical in terms of memory, since it only needs access to Hessian-vector products (section 2.3). A high-level view is illustrated in Algorithm 2, where CG stands for one step of conjugate-gradient (a stopping condition, line search and some intermediate variables were omitted for clarity). Note that for every update of $w$ (outer loop), Algorithm 2 must perform several steps of CG (inner loop) to find a *single* search direction $z$.

We propose a number of changes in order to eliminate this costly inner loop. The first is to reuse the previous search direction $z$ to warm-start the inner iterations, instead of resetting $z$ each time (Algorithm 2, line 2). If $z$ does not change abruptly, then this should help reduce the number of CG iterations, by starting closer to the solution. The second change is to use this fact to dramatically reduce the inner loop iterations to just one ($R = 1$). A different interpretation is that we now *interleave* the updates of the search direction $z$ and parameters $w$ (Algorithm 1, lines 4 and 5), instead of nesting them (Algorithm 2, lines 4 and 6).

Unfortunately, this change loses the guarantees of the CG method, which depend on the starting point being $z_0 = -J(w_t)$ (Wright & Nocedal, 1999, p. 124). This loss of guarantee was verified in practice, as we found the resulting algorithm extremely unstable. Our third change is then to replace CG with gradient descent, which has no such dependency. Differentiating eq. 14 w.r.t. $z$ yields:

$$\triangle_z = J_{\hat{f}(z)} = \hat{H}z + J \tag{15}$$

Applying these changes to the Hessian-free method (Algorithm 2) results in Algorithm 1. By contrasting it to momentum GD (eq. 2-3), we can see that it is equivalent, except for an extra curvature term $\hat{H}(w)z$. In order to establish this equivalence, we introduced a factor $\rho$ that decays $z$ each step (Algorithm 1, line 4), whereas a pure gradient descent step on $z$ would not include this factor. We can obtain it formally by simply regularizing the quadratic model in eq. 13 with the term $(1 - \rho) \|z\|^2$, which is small when $\rho \simeq 1$ (the recommended setting for the momentum parameter (He et al., 2016)). Due to the addition of curvature to momentum GD, which is also known as the heavy-ball method, we name our algorithm CURVEBALL.

**Implementation.** Using the fast Hessian-vector products from section 2.3, it is easy to implement eq. 15, including a regularization term $\lambda I$ (section 2.1). We can further improve eq. 15 by grouping

Table 1: **Optimiser comparison on small degenerate datasets.** For each optimiser, we report the mean$\pm$standard deviation of the number of iterates taken to reach the solution. For the stochastic Rosenbrock function, $\mathcal{U}[\lambda_1, \lambda_2]$ denotes noise drawn from $\mathcal{U}[\lambda_1, \lambda_2]$ (see Sec. 4 for details)

| | Deterministic | Rosenbrock $\mathcal{U}[0, 1]$ | $\mathcal{U}[0, 3]$ | Raihimi & Recht |
|---|---|---|---|---|
| SGD + momentum | $370 \pm 40$ | $846 \pm 217$ | $4069 \pm 565$ | $95 \pm 2$ |
| Adam (Kingma & Ba, 2014) | $799 \pm 160.5$ | $1290 \pm 476$ | $2750 \pm 257$ | $95 \pm 5$ |
| Levenberg-Marquardt (Moré, 1978) | $16 \pm 4$ | $14 \pm 3$ | $17 \pm 4$ | $9 \pm 4$ |
| BFGS (Wright & Nocedal, 1999, p. 136) | $19 \pm 4$ | $44 \pm 21$ | $63 \pm 29$ | $43 \pm 21$ |
| Exact Hessian | $14 \pm 1$ | $\mathbf{10 \pm 3}$ | $17 \pm 4$ | $\mathbf{9 \pm 0.5}$ |
| CURVEBALL (proposed) | $\mathbf{13 \pm 0.5}$ | $12 \pm 1$ | $\mathbf{13 \pm 1}$ | $35 \pm 11$ |

the operations to minimize the number of automatic differentiation (back-propagation) steps:

$$\triangle_z = \left( J_\phi H_L J_\phi^T + \lambda I \right) z + J_\phi J_L \tag{16}$$

$$= J_\phi \left( H_L J_\phi^T z + J_L \right) + \lambda z \tag{17}$$

In this way, the total number of passes over the model is two: we compute $J_\phi v$ and $J_\phi^T v'$ products, implemented respectively as one RMAD (back-propagation) and one FMAD operation (section 2.2).

**Automatic $\rho$ and $\beta$ hyper-parameters in closed form.** Our proposed method introduces a few hyper-parameters, which just like with SGD, would require tuning for different settings. Ideally, we would like to have no tuning at all. Fortunately, the quadratic minimization interpretation in eq. 14 allows us to draw on standard results in optimisation. At any given step, the optimal $\rho$ and $\beta$ can be obtained by solving a $2 \times 2$ linear system (Martens & Grosse, 2015, sec. 7):

$$\left[ \begin{array}{c} \beta \\ -\rho \end{array} \right] = \left[ \begin{array}{cc} \Delta_z^T \hat{H} \Delta_z & z^T \hat{H} \Delta_z \\ z^T \hat{H} \Delta_z & z^T \hat{H} z \end{array} \right]^{-1} \left[ \begin{array}{c} J^T \Delta_z \\ J^T z \end{array} \right] \tag{18}$$

Note that, in calculating the proposed update (eq. 16), the quantities $\Delta_z$, $J_\phi^T z$ and $J_L$ have already been computed and can now be reused. Together with the fact that $\hat{H} = J_\phi H_L J_\phi^T$, this means that the elements of the above $2 \times 2$ matrix can be computed with only one additional forward pass.

**Automatic $\lambda$ hyper-parameter rescaling.** The regularization term $\lambda I$ (eq. 4) can be interpreted as a trust-region (Wright & Nocedal, 1999, p. 68). When the second-order approximation holds well, $\lambda$ can be small, corresponding to an unregularized Hessian and a large trust-region. Conversely, a poor fit requires a correspondingly large $\lambda$. We can measure the difference (or ratio) between the objective change predicted by the quadratic fit ($\hat{f}$) and the real objective change ($f$), by computing $\gamma = (f(w + z) - f(w)) / \hat{f}(z)$. This requires one additional evaluation of the objective for $f(w + z)$, but otherwise relies only on previously computed quantities. This makes it a very attractive estimate of the trust region, with $\gamma = 1$ corresponding to a perfect approximation. Following (Wright & Nocedal, 1999, p. 69), we evaluate $\gamma$ every 5 iterations, decreasing $\lambda$ by a factor of 0.999 when $\gamma > 3/2$, and increasing by the inverse factor when $\gamma < 1/2$. We noted that our algorithm is not very sensitive to the initial $\lambda$. In experiments using batch-normalization (section 4), we simply initialize it to one, otherwise setting it to 10. We plot the evolution of automatically tuned hyper-parameters in fig. 8 (Appendix B).

**Convergence.** In addition to the usual absence of strong guarantees for non-convex problems, which applies in our setting (deep neural networks), there is an added difficulty due to the recursive nature of our algorithm (the interleaved $w$ and $z$ steps). Our method is a variant of the heavy-ball method (Lessard et al., 2016; Flammarion & Bach, 2015) (by adding a curvature term), which until very recently had resisted establishing global convergence rates that improve on gradient descent without momentum (Loizou & Richtárik (2017), table 1), and even then only for strongly convex or quadratic functions.

For this reason, we present proofs for two more tractable cases (Appendix A). The first is the global linear convergence of our method for convex quadratic functions, which allows a direct inspection

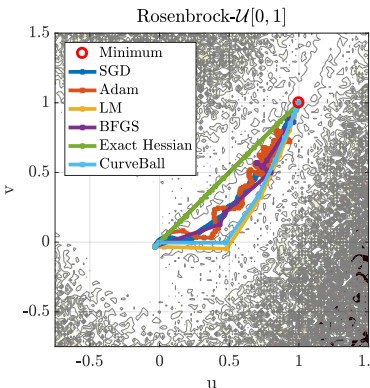 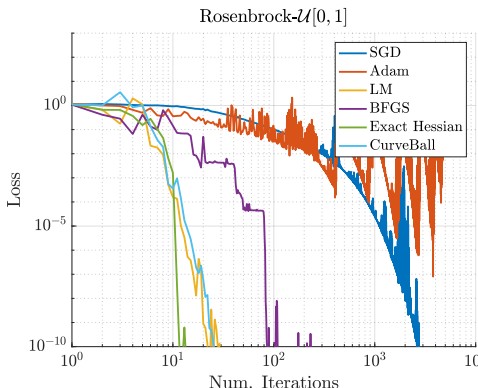

Figure 1: **Degenerate optimisation problems with known solutions**. Left: Trajectories on the Stochastic Rosenbrock function for different solvers (darker shaded regions denote higher function values). Right: evolution of the loss per iterations for the plotted trajectories.

of the region of convergence as well as its rate (Theorem A.1). The second establishes that, for convex non-quadratic functions, CURVEBALL's steps are always in a descent direction, when using the automatic hyper-parameter tuning of eq. 18 (Theorem A.2). We note that in practice, due to the Gauss-Newton approximation and the trust region (eq. 4), the effective Hessian is guaranteed to be positive-definite.

Similarly to momentum SGD, our main claim as to the method's suitability for non-convex deep network optimisation is necessarily empirical, based on the extensive experiments in section 4, which show strong performance on several large-scale problems with no hyper-parameter tuning.

## 4 EXPERIMENTS

**Degenerate problems with known solutions.** While the main purpose of our optimizer is its application to large-scale deep learning architecture, we begin by applying our method to problems of limited complexity, with the goal of exploring the strengths and weaknesses of our approach in an interpretable domain. We perform a comparison with two popular first order solvers — SGD with momentum and Adam (Kingma & Ba, 2014)[3], as well as with more traditional methods such as Levenberg-Marquardt, BFGS (Wright & Nocedal, 1999, p. 136) (with cubic line-search) and Newton's method with the exact Hessian. The first problem we consider is the search for the minimum of the two-dimensional Rosenbrock test function, which has the useful benefit of enabling us to visualise the trajectories found by each optimiser. Specifically, we use the stochastic variant of this function (Yang & Deb, 2010), $\mathcal{R} : \mathbb{R}^2 \to \mathbb{R}$:

$$\mathcal{R}(u, v) = (1 - u)^2 + 100\epsilon_i(v - u^2)^2, \tag{19}$$

where at each evaluation of the function, a noise sample $\epsilon_i$ is drawn from a uniform distribution $\mathcal{U}[\lambda_1, \lambda_2]$ with $\lambda_1, \lambda_2 \in \mathbb{R}$ (we can recover the deterministic Rosenbrock function with $\lambda_1 = \lambda_2 = 1$). To assess robustness to noise, we compare each optimiser on the deterministic formulation and two stochastic variants (with differing noise regimes). We also consider a second problem of interest, recently introduced by Rahimi & Recht (2017). It consists of fitting a deep network with only two linear layers to a dataset where sample inputs $x$ are related to sample outputs $y$ by the relation $y = Ax$, where $A$ is an ill-conditioned matrix (with condition number $\epsilon = 10^5$).

The results are shown in Table 1. We use a grid-search to determine the best hyper-parameters for both SGD and Adam (reported in appendix B.1). We report the number of iterates taken to reach the solution, with a tolerance of $\tau = 10^{-4}$. Statistics are computed over 100 runs of each optimiser. We observe that first-order methods perform poorly in all cases, and moreover show a very high

---

[3]We also experimented with RMSProp (Tieleman & Hinton, 2012), AdaGrad (Duchi et al., 2011) and AdaDelta (Zeiler, 2012), but found that these methods consistently underperformed Adam and SGD on these "toy" problems.

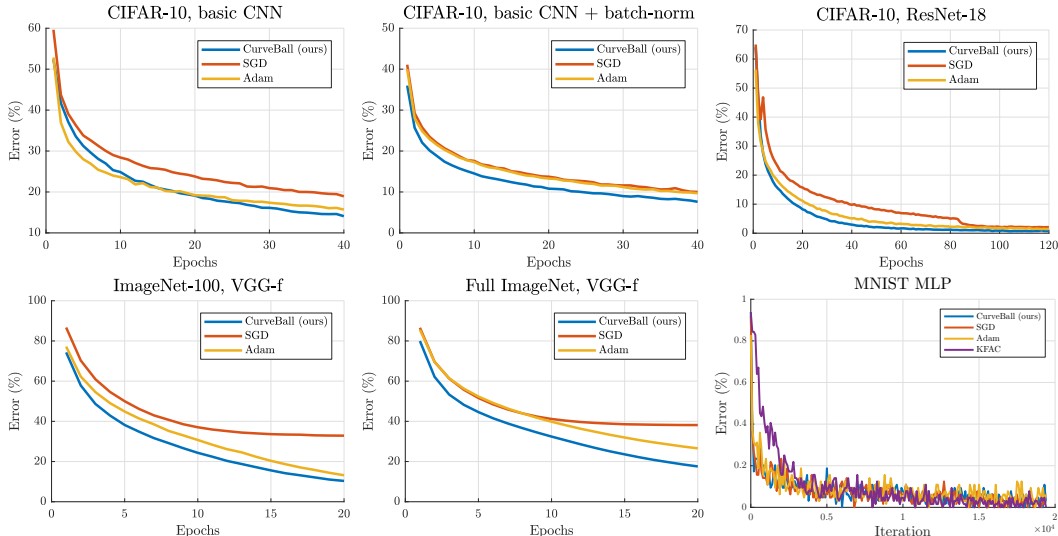

Figure 2: **Comparison with the different optimisers for various datasets and networks**. The evolution of the training error is shown, as it is the quantity being optimised. CURVEBALL performs well under a variety of realistic settings, including large-scale datasets (ImageNet), the presence of batch-normalization, and severely over-parameterised models (ResNet).

variance of results. The Newton method with an exact Hessian[4] generally performs best, followed closely by Levenberg-Marquardt (LM), however they are impractical for larger-scale problems. Our method delivers comparable (and sometimes better) performance despite avoiding a costly Hessian inversion. On the other hand, the performance of BFGS, which approximates the Hessian with a buffer of parameter updates, seems to correlate negatively with the level of noise.

Fig. 1 shows example trajectories. The slow, oscillating behaviour of first-order methods is noticeable, as well as the impact of noise on the BFGS steps. On the other hand, CURVEBALL, Newton and LM converge in few iterations.

**CIFAR.** We now turn to the task of training deep networks on more realistic datasets. Second-order methods are typically not used in such scenarios, due to the large number of parameters and stochastic sampling. We start with a basic 5-layer convolutional neural network (CNN).[5] We train this network for 20 epochs on CIFAR-10, with and without batch-normalization (which is known to improve conditioning (Kohler et al., 2018)) using for every experiment a mini-batch size of 128. To assess optimiser performance on larger models, we also train a much larger ResNet-18 model (He et al., 2016). As baselines, we picked SGD (with momentum) and Adam, which we found to outperform the competing first-order optimisers. Their learning rates are chosen from the set $10^{-k}$, $k \in \mathbb{N}$ with a grid search for the basic CNN, while for the ResNet SGD uses the schedule recommended by the authors (He et al., 2016). We focus on the training error, since it is the quantity being optimised by eq. 1 (validation error is discussed below). The results can be seen in Fig. 2 (top row). We observe that in each setting, CURVEBALL outperforms its competitors, in a manner that is robust to normalisation and model type.

**ImageNet.** To assess the practicality of our method at larger scales, we apply it to the classification task on the large-scale ImageNet dataset. We report results of training on both a medium-scale setting using a subset formed from the images of 100 randomly sampled classes as well as the large-scale setting, by training on the full dataset. Both experiments use the VGG-f architecture with mini-batch size of 256 and follow the settings described by Chatfield et al. (2014). The results are depicted in Fig. 2. We see that our method provides compelling performance against popular first order solvers in both cases, and that interestingly, its margin of improvement grows with the scale of the dataset.

---

[4]When the Hessian contains negative eigenvalues, their absolute values are used (Dauphin et al., 2014).

[5]The basic CNN has $5 \times 5$ filters and ReLU activations, and $3 \times 3$ max-pooling layers (with stride 2) after each of the first 3 convolutions. The number of output channels are, respectively, $(32, 32, 64, 64, 10)$.

Table 2: Best error in percentage (training/validation) for different models and optimisation methods. CURVEBALL $\lambda$ denotes use of $\lambda$ rescaling (sec. 3). Numbers in bracket show validation error with additional dropout regularisation (rate 0.3). The first three columns are trained on CIFAR-10, while the fourth is trained on ImageNet-100.

| Model | Basic | Basic + BN | ResNet-18 | VGG-f |
|---|---|---|---|---|
| CURVEBALL $\lambda$ | **14.1** / 19.9 | **7.6** / 16.3 | **0.7** / 15.3 (13.5) | **10.3** / **33.5** |
| CURVEBALL | 15.3 / **19.3** | 9.4 / **15.8** | 1.3 / 16.1 | 12.7 / 33.8 |
| SGD | 18.9 / 21.1 | 10.0 / 16.1 | 2.1 / **12.8** | 32.9 / 41.7 |
| Adam | 15.7 / 19.7 | 9.6 / 16.1 | 1.4 / 14.0 | 13.2 / 35.9 |

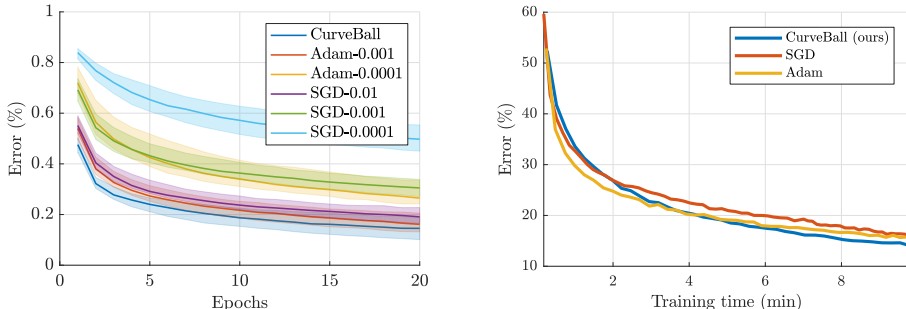

Figure 3: (Left) **Results of 50 randomly-generated architectures on CIFAR10.** The median (thick lines) and 25[th]-75[th] percentiles (shaded region) are shown. Numbers in the legend represent learning rates (fixed for CURVEBALL). (Right) **Training error vs. wall clock time** (basic CIFAR-10 model).

**Comparison to other second-order methods on MNIST.** In order to compare ours with existing second-order methods, we use the public KFAC (Martens & Grosse, 2015) implementation made available by the authors and run a simple experiment on the MNIST dataset. In this scenario a four layer MLP (with output sizes 128-64-32-10) with hyperbolic tangent activations is trained on this classification task. We closely follow the same protocol as Martens & Grosse (2015) for layer initialisation and data normalisation, with batch size 64. We show results in Fig 2 (bottom row, right) with the best learning rate for each method. On this problem our method performs comparably to first order solvers, while KFAC makes less progress until it has stabilised its Fisher matrix estimation.

**Random architecture results.** It can be argued that standard architectures are biased to favour SGD, since it was used in the architecture searches, and architectures in which it failed to optimise were discarded (He et al., 2016). It would be useful to assess the optimisers' ability to generalise across architectures, testing how well they perform regardless of the network model. We make an attempt in this direction by comparing the optimisers on 50 deep CNN architectures that are generated randomly (see appendix B.3 for details). In addition to being more architecture-agnostic, this makes any hand-tuning of hyper-parameters infeasible, which we believe to be a fair requirement for a dependable optimiser. The results on CIFAR10 are shown in figure 3 (left), as the median across all runs (thick lines) and 25[th]-75[th] percentiles (shaded regions). CURVEBALL consistently outperforms first-order methods, with the bulk of the achieved errors below those of SGD and Adam.

**Wall-clock time.** To provide an estimate of the relative efficiency of each model, Fig. 3 shows wall clock time on the basic CIFAR-10 model (without batch norm). Importantly, from a practical perspective, we observe that our method is competitive with first order solvers, while not requiring any tuning. Moreover, our prototype implementation includes custom FMAD operations which have not received the same degree of optimisation as RMAD (back-propagation), and could further benefit from careful engineering. We also experimented with a Hessian-free optimiser (based on conjugate gradients) (Martens, 2010). We show a comparison in logarithmic time in the appendix (Fig. 7). Due to the costly CG operation, which requires several passes through the network, it is an order of magnitude slower than the first-order methods and our own second-order method. This validates our initial motivation of designing a Hessian-free method without the inner CG loop (Section 3).

**Overfitting and validation error**    While the focus of this work is optimisation, it is also of interest to compare the validation errors attained by the trained models – these are reported Table 2. We observe that models trained with the proposed method exhibit better training and validation error on most models, with the exception of ResNet where overfitting plays a more significant role. However, we note that this could be addressed with better regularisation, and we show one such example, by also reporting the validation error with a dropout rate of 0.3 in brackets.

## 5    RELATED WORK

While second order methods have proved to be highly effective tools for optimising deterministic functions (Moré, 1978; Wright & Nocedal, 1999, p. 164) their application to stochastic optimisation, and in particular to deep neural networks remains an active area of research. Many methods have been developed to improve stochastic optimisation with curvature information to avoid slow progress in ill-conditioned regions (Dauphin et al., 2014), while avoiding the cost of storing and inverting a Hessian matrix. A popular approach is to construct updates from a buffer of parameter gradients and their first-and-second-order moments at previous iterates (*e.g.* AdaGrad (Duchi et al., 2011), AdaDelta (Zeiler, 2012), RMSProp (Tieleman & Hinton, 2012) or Adam (Kingma & Ba, 2014)). These solvers benefit from needing no additional function evaluations beyond traditional mini-batch stochastic gradient descent. Typically they set adaptive learning rates by making use of empirical estimates of the curvature with a *diagonal* approximation to the Hessian (*e.g.* Zeiler (2012)) or a rescaled diagonal Gauss-Newton approximation (*e.g.* Duchi et al. (2011)). While the diagonal structure decreases the computational cost, their overall efficiency remains limited and in many cases can be matched by a well tuned SGD solver (Wilson et al., 2017).

Second order solvers take a different approach, investing more computation per iteration in the hope of achieving higher quality updates. Trust-region methods (Conn et al., 2000) and cubic regularization (Nesterov & Polyak, 2006; Cartis et al., 2011) are canonical examples. To achieve this higher quality, they invert the Hessian matrix $H$, or a tractable approximation such as the Gauss-Newton approximation (Martens, 2010; Martens & Sutskever, 2011; Botev et al., 2017) (described in section 2), or other regularized (Dauphin et al., 2014) or subsampled versions of the Hessian (Kohler & Lucchi, 2017; Xu et al., 2017). Another line of work belonging to the trust-region family (Conn et al., 2000), which has proven effective for tasks such as classification, introduces second order information with *natural gradients* (Amari, 1998). In this context, it is common to derive a loss function from a Kullback-Leibler (KL) divergence. The natural gradient makes use of the infinitesimal distance induced by the latter to follow the curvature in the Riemannian manifold equipped with this new distance. In practice the natural gradient method amounts to replacing the Hessian $H$ in the modified gradient formula $H^{-1}J$ with the Fisher matrix $F$, which facilitates traversal of the optimal path in the metric space induced by the KL-divergence. Since the seminal work of Amari (1998) several authors have studied variations of this idea. TONGA (Roux et al., 2008) relies on the empirical Fisher matrix where the previous expectation over the model predictive distribution is replaced by the sample predictive distribution. The works of Pascanu & Bengio (2013) and Martens established a link between Gauss-Newton methods and the natural gradient. More recently Martens & Grosse (2015) introduced the KFAC optimiser which uses a block diagonal approximation of the Fisher matrix. This was shown to be an efficient stochastic solver in several settings, but it remains a computationally challenging approach for larger-scale deep networks problems.

Many of the methods discussed above perform an explicit system inversion that can often prove prohibitively expensive (Yang & Amari, 1998). Consequently, a number of works (Martens, 2010; Martens & Sutskever, 2011; Zhang et al., 2017) have sought to exploit the cheaper computation of Hessian-vector products via automatic differentiation (Pearlmutter, 1994; Schraudolph, 2002), to perform system inversions with conjugate gradients (Hessian-free methods). Other approaches (Carmon et al., 2018; Agarwal et al., 2017) have resorted to rank-1 approximations of the Hessian for efficiency. While these methods have had some success, they have only been demonstrated on single-layer models of moderate scale compared to the state-of-the-art in deep learning. We speculate that the main reason they are not widely adopted is their requirement of several steps (network passes) per parameter update (Sutskever et al., 2013; Koh & Liang, 2017), which would put them at a similar disadvantage w.r.t. first-order methods as the Hessian-free method that we tested (fig. 7 in the appendix). Perhaps more closely related to our approach, Orr (1995) uses automatic differentiation to compute Hessian-vector products to construct adaptive, per-parameter learning rates.

The closest method is LiSSA (Agarwal et al., 2017), which is built around the idea of approximating the Hessian inverse with a Taylor series expansion. This series can be implemented as the recursion $H_{(r)}^{-1} = I + (I - H)H_{(r-1)}^{-1}$, starting with $H_{(0)}^{-1} = I$. Since LiSSA is a type of Hessian-free method, the core of the algorithm is similar to Algorithm 2: it also refines an estimate of the Newton step iteratively, but with a different update rule in line 4. With some simple algebraic manipulations, we can use the Taylor recursion to write this update in a form that is similar to ours: $z_{r+1} = z_r - \hat{H}z_r - J$. This looks similar to our gradient-descent-based update with a learning rate of $\beta = 1$ (Alg. 1, lines 3-4), with some key differences. First, they reset the state of the step estimate for every mini-batch (Alg. 2, line 2). Reusing past solutions, like momentum-SGD, is an important factor in the performance of our algorithm, since we only have to perform one update per mini-batch. In contrast, Agarwal et al. (2017) report a typical number of inner-loop updates ($R$ in Alg. 2) equal to the number of samples (e.g. $R = 10,000$ for a tested subset of MNIST). While this is not a problem for their tested case of linear Support Vector Machines, since each update only requires one inner-product, the same does not apply to deep neural networks. Second, they invert the Hessian independently for each mini-batch, while our method aggregates the implicit Hessian across all past mini-batches (with a forgetting factor of $\rho$). Since batch sizes are orders of magnitude smaller than the number of parameters (e.g. 256 samples vs. 60 million parameters for the VGG-f), the Hessian matrix for a mini-batch is a poor substitute for the Hessian of the full dataset in these problems, and severely ill-conditioned. Third, while their method fixes $\beta$ to 1, we found that setting it correctly can be used to attain convergence on ill-conditioned problems. For example, even on quadratic problems, we show that this parameter needs to be carefully chosen to avoid divergence (Theorem A.1). The gradient descent interpretation used in our work, contrasted to the Taylor series recursion of Agarwal et al. (2017), thus brings an additional degree of freedom that may be useful to relax other assumptions. Finally, while they demonstrate improved performance on convex problems with linear models, we focus on the needs of training deep networks on large datasets (millions of samples and parameters), on which no previous Newton method has been able to surpass the first-order methods that are commonly used by the deep learning community.

## 6 Conclusions and future work

In this work, we have proposed a practical second-order solver that has been specifically tailored for deep-learning-scale stochastic optimisation problems. We showed that our optimiser can be applied to a large range of datasets and reach better training error than first order method with the same number of iterations, with essentially no hyper-parameters tuning. In future work, we intend to bring more improvements to the wall-clock time of our method by engineering the FMAD operation to the same standard as back-propagation, and study optimal trust-region strategies to obtain $\lambda$ in closed-form.

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

# A    ADDITIONAL PROOFS

## A.1    DERIVATION OF AUTOMATIC HYPER-PARAMETER TUNING IN CLOSED-FORM

We rewrite the problem in eq. 14 as a minimization over $\rho$ and $\beta$ where $z' = \rho z - \beta \Delta z$:

$$z = \arg\min_{\rho, \beta} \hat{f}(z') \tag{20}$$

$$= \arg\min_{\rho, \beta} \begin{bmatrix} \rho \\ -\beta \end{bmatrix}^T [\ z \quad \Delta_z\ ]^T J + \frac{1}{2} \begin{bmatrix} \rho \\ -\beta \end{bmatrix}^T [\ z \quad \Delta_z\ ]^T H [\ z \quad \Delta_z\ ] \begin{bmatrix} \rho \\ -\beta \end{bmatrix} \tag{21}$$

$$= \arg\min_{\rho, \beta} \begin{bmatrix} \rho \\ -\beta \end{bmatrix}^T \begin{bmatrix} z^T J \\ \Delta_z^T J \end{bmatrix} + \frac{1}{2} \begin{bmatrix} \rho \\ -\beta \end{bmatrix}^T \begin{bmatrix} z^T \hat{H} z & z^T \hat{H} \Delta_z \\ z^T \hat{H} \Delta_z & \Delta_z^T \hat{H} \Delta_z \end{bmatrix} \begin{bmatrix} \rho \\ -\beta \end{bmatrix}. \tag{22}$$

Since $\hat{f}$ is a quadratic function of $\rho$ and $\beta$ with PSD Hessian it is therefore convex and we can find its extrema by cancelling the gradient:

$$\nabla_{\rho, \beta} \hat{f}(z') = 0. \tag{23}$$

Therefore, we have:

$$\begin{bmatrix} z^T J \\ -\Delta_z^T J \end{bmatrix} + \begin{bmatrix} z^T \hat{H} z & -z^T \hat{H} \Delta_z \\ -z^T \hat{H} \Delta_z & \Delta_z^T \hat{H} \Delta_z \end{bmatrix} \begin{bmatrix} \rho \\ \beta \end{bmatrix} = 0 \tag{24}$$

$$\begin{bmatrix} -\rho \\ \beta \end{bmatrix} = \begin{bmatrix} z^T \hat{H} z & z^T \hat{H} \Delta_z \\ z^T \hat{H} \Delta_z & \Delta_z^T \hat{H} \Delta_z \end{bmatrix}^{-1} \begin{bmatrix} z^T J \\ \Delta_z^T J \end{bmatrix}, \tag{25}$$

where the last equality can be computed by inverting the $2 \times 2$ matrix explicitly.

## A.2    PROOF OF CONVERGENCE IN THE QUADRATIC CASE

**Theorem A.1.** *Let $f$ be a convex quadratic function, and its hyper-parameters $\beta > 0$, $\rho > 0$ satisfy*

$$\tfrac{3}{2} \beta h_{\max} - 1 < \rho < 1 + \beta h_{\min}, \tag{26}$$

*where $h_{\min}$ and $h_{\max}$ are the smallest and largest eigenvalues of the Hessian $H$, respectively. Then Algorithm 1 converges linearly to the minimum of $f$.*

**Corollary A.1.1.** *Algorithm 1 converges for any momentum parameter $0 < \rho < 1$ with a sufficiently small learning rate $\beta > 0$, regardless of the (PSD) Hessian spectrum.*

*Proof of Theorem A.1.* We follow similar derivations on quadratic models by previous work on the heavy-ball method (Goh, 2017; Lessard et al., 2016; Flammarion & Bach, 2015), but including our curvature term in the update. We assume the quadratic model:

$$f(w) = \frac{1}{2} w^T H w - b^T w, \tag{27}$$

which has Hessian matrix $H$, and gradient $J(w) = Hw - b$.

Without loss of generality, we will consider the pure Newton method, where $H$ is not regularized $(\lambda = 0)$:[6]

$$z_{t+1} = \rho z_t - \beta(H z_t + J(w_t)) \tag{28}$$
$$w_{t+1} = w_t + z_{t+1} \tag{29}$$

Eq. 28 can be rearranged to

$$z_{t+1} = (\rho I - \beta H) z_t - \beta J(w_t). \tag{30}$$

---

[6]For the general case, the momentum parameter $\rho$ is simply replaced by the slightly perturbed value $\rho - \beta \lambda$ (since $\rho \gg \beta \lambda$), and similar derivations follow.

We now perform a change of variables to diagonalize the Hessian, $H = Q\text{diag}(h)Q^T$, with $Q$ orthogonal and $h$ the vector of eigenvalues. Let $w^* = \arg\min_w f(w) = H^{-1}b$ be the optimal solution of the minimization. Then, replacing $w_t = Qx_t + w^*$ in eq. 30:

$$Qy_{t+1} = (\rho I - \beta H)Qy_t - \beta H Qx_t \tag{31}$$

with $J = H(Qx_t + w^*) - b = H(Qx_t + H^{-1}b) - b = HQx_t$.

Then, expanding $H$ with its eigendecomposition,

$$Qy_{t+1} = \rho Qy_t - \beta Q\text{diag}(h)Q^T Qy_t - \beta Q\text{diag}(h)Q^T Qx_t. \tag{32}$$

Left-multiplying by $Q^T$,and canceling out $Q$ due to orthogonality,

$$y_{t+1} = \rho y_t - \beta\text{diag}(h)y_t - \beta\text{diag}(h)x_t. \tag{33}$$

Similarly for eq. 29, replacing $z_t = Qy_t$ yields

$$x_{t+1} = x_t + y_{t+1}. \tag{34}$$

Note that each pair formed by the corresponding element of $y_t$ and $x_t$ is an independent system with only 2 variables, since the pairs do not interact (eq. 33 and 34 only contain element-wise operations). From now on, we will be working on the $i$th element of each vector.

We can thus write eq. 33 and 34 (for a single element $i$ of each) as a vector equation:

$$\begin{bmatrix} 1 & 0 \\ -1 & 1 \end{bmatrix}\begin{bmatrix} y_{t+1,i} \\ x_{t+1,i} \end{bmatrix} = \begin{bmatrix} \rho - \beta h_i & -\beta h_i \\ 0 & 1 \end{bmatrix}\begin{bmatrix} y_{t,i} \\ x_{t,i} \end{bmatrix}. \tag{35}$$

The matrix on the left is necessary to express the fact that the $y_{t+1}$ factor in eq. 34 must be moved to the left-hand side, which corresponds to iteration $t + 1$ ($x_{t+1} - y_{t+1} = x_t$). Left-multiplying eq. 35 by the inverse,[7]

$$\begin{bmatrix} y_{t+1,i} \\ x_{t+1,i} \end{bmatrix} = \begin{bmatrix} \rho - \beta h_i & -\beta h_i \\ \rho - \beta h_i & 1 - \beta h_i \end{bmatrix}\begin{bmatrix} y_{t,i} \\ x_{t,i} \end{bmatrix}. \tag{36}$$

This is the transition matrix $R_i$ that characterizes the iteration, and taking its power models multiple iterations in closed-form:

$$\begin{bmatrix} y_{t,i} \\ x_{t,i} \end{bmatrix} = R_i^t \begin{bmatrix} y_{0,i} \\ x_{0,i} \end{bmatrix}. \tag{37}$$

The two eigenvalues of $R_i$ are given in closed-form by:

$$\text{eig}(R_i) = \frac{1}{2}\left(\rho - 2\beta h_i + 1 \pm \sqrt{(\rho - 2\beta h_i)^2 - 2\rho + 1}\right) \tag{38}$$

The series in eq. 37 converges when $|\text{eig}(R_i)| < 1$ simultaneously for both eigenvalues, which is equivalent to:

$$\tfrac{3}{2}\beta h_i - 1 < \rho < 1 + \beta h_i, \tag{39}$$

with $\rho > 0$ and $\beta h_i > 0$. Note that when using the Gauss-Newton approximation of the Hessian, $h_i > 0$ and thus the last condition simplifies to $\beta > 0$.

Since eq. 39 has to be satisfied for every eigenvalue, we have

$$\tfrac{3}{2}\beta h_{\max} - 1 < \rho < 1 + \beta h_{\min}, \tag{40}$$

with $h_{\min}$ and $h_{\max}$ the smallest and largest eigenvalues of the Hessian $H$, respectively, proving the result.

The rate of convergence is the largest of the two values $|\text{eig}(R_i)|$. When the argument of the square root in eq. 38 is non-negative, it does not admit an easy interpretation; however, when it is negative, eq. 38 simplifies to:

$$|\text{eig}(R_i)| = \sqrt{\rho - \beta h_i}. \tag{41}$$

$\square$

---

[7]We have: $\begin{bmatrix} 1 & 0 \\ -1 & 1 \end{bmatrix}^{-1} = \begin{bmatrix} 1 & 0 \\ 1 & 1 \end{bmatrix}$

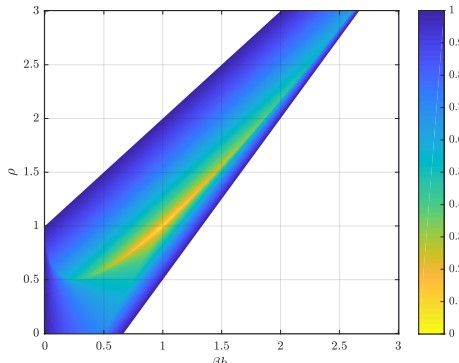

Figure 4: Convergence rate as a function of hyper-parameters $\rho$, $\beta$, and Hessian eigenvalue $h_i$. Lower values (brighter) are better. The white areas show regions of non-convergence.

### A.2.1 GRAPHICAL INTERPRETATION

The convergence rate for a single eigenvalue is illustrated in Figure 4. Graphically, the regions of convergence for different eigenvalues will differ only by a scale factor along the $\beta h_i$ axis (horizontal stretching of Figure 4). Moreover, the largest possible range of $\beta h_i$ values is obtained when $\rho = 1$, and that range is $0 < \beta h_i < \frac{4}{3}$. We can infer that the intersection of the regions of convergence for several eigenvalues will be maximized with $\rho = 1$, for any fixed $\beta$.

### A.3 PROOF OF GUARANTEED DESCENT ON GENERAL NON-CONVEX FUNCTIONS

**Theorem A.2.** *Let the Hessian $\hat{H}_{t+1}$ be positive definite (which holds when the objective is convex or when Gauss-Newton approximation and trust region are used). Then the update $z_{t+1}$ in Algorithm 1 is a descent direction when $\beta$ and $\rho$ are chosen according to eq. 18, and $z_{t+1} \neq 0$.*

*Proof.* To show that the update represents a descent direction, it suffices to show that $J^T z_{t+1} < 0$ (where we have written $J = J(w_t)$ to simplify notation). Since the surrogate Hessian $\hat{H}_{t+1}$ is positive definite (PD) by construction, the update $z_{t+1} = \rho z_t - \beta \Delta_{z_{t+1}}$ satisfies $z_{t+1}^T \hat{H}_{t+1} z_{t+1} > 0$. It is therefore sufficient to prove that $J^T z_{t+1} + z_{t+1}^T \hat{H}_{t+1} z_{t+1} \leq 0$.

It follows from their definition in eq. (18) that $\rho$ and $\beta$ minimise the RHS of

$$
J^T z_{t+1} + \frac{1}{2} z_{t+1}^T \hat{H}_{t+1} z_{t+1} =
$$
$$
\left[ \begin{array}{c} J^T \Delta_{z_{t+1}} \\ J^T z_t \end{array} \right]^T \left[ \begin{array}{c} -\beta \\ \rho \end{array} \right] + \frac{1}{2} \left[ \begin{array}{c} -\beta \\ \rho \end{array} \right]^T \left[ \begin{array}{cc} \Delta_{z_{t+1}}^T \hat{H}_{t+1} \Delta_{z_{t+1}} & z_t^T \hat{H}_{t+1} \Delta_{z_{t+1}} \\ z_t^T \hat{H}_{t+1} \Delta_{z_{t+1}} & z_t^T \hat{H}_{t+1} z_t \end{array} \right] \left[ \begin{array}{c} -\beta \\ \rho \end{array} \right] \quad (42)
$$

In particular, they minimise a quadratic form in $(-\beta, \rho)$ with the following symmetric Hessian

$$
K = \left[ \begin{array}{cc} \Delta_{z_{t+1}}^T \hat{H}_{t+1} \Delta_{z_{t+1}} & z_t^T \hat{H}_{t+1} \Delta_{z_{t+1}} \\ z_t^T \hat{H} \Delta_{z_{t+1}} & z_t^T \hat{H}_{t+1} z_t \end{array} \right]. \quad (43)
$$

Moreover, for any $x = (x_1, x_2) \in \mathbb{R}^2$,

$$
x^T K x = \left[ \begin{array}{c} x_1 \\ x_2 \end{array} \right]^T \left[ \begin{array}{cc} \Delta_{z_{t+1}}^T \hat{H}_{t+1} \Delta_{z_{t+1}} & z_t^T \hat{H}_{t+1} \Delta_{z_{t+1}} \\ z_t^T \hat{H} \Delta_{z_{t+1}} & z_t^T \hat{H}_{t+1} z_t \end{array} \right] \left[ \begin{array}{c} x_1 \\ x_2 \end{array} \right]
$$
$$
= (x_1 \Delta_{z_{t+1}} + x_2 z_t)^T \hat{H} (x_1 \Delta_{z_{t+1}} + x_2 z_t). \quad (44)
$$

Consequently, K is guaranteed to be Positive Semidefinite (PSD) and the form is convex with zero gradient at the minimum. Since $z_{t+1} \neq 0$, it follows that at least one of the following holds: (1) K is

invertible and hence PD (rather than simply PSD); (2) one of factors $z_t = 0$ or $\Delta_{z_{t+1}} = 0$ is zero; (3) the factors $z_t = 0$ and $\Delta_{z_{t+1}} = 0$ are colinear. In the first case we have,

$$
\begin{aligned}
J^T z_{t+1} + \frac{1}{2} z_{t+1}^T \hat{H}_{t+1} z_{t+1} = \\
-\frac{1}{2} \left[ \begin{array}{c} J^T \Delta_{z_{t+1}} \\ J^T z_t \end{array} \right]^T \left[ \begin{array}{cc} \Delta_{z_{t+1}}^T \hat{H}_{t+1} \Delta_{z_{t+1}} & z_t^T \hat{H}_{t+1} \Delta_{z_{t+1}} \\ z_t^T \hat{H}_{t+1} \Delta_{z_{t+1}} & z_t^T \hat{H}_{t+1} z_t \end{array} \right]^{-1} \left[ \begin{array}{c} J^T \Delta_{z_{t+1}} \\ J^T z_t \end{array} \right]
\end{aligned} \tag{45}
$$

Since the inverse of a PD matrix is PD, the RHS of eq. (42) is negative. Further, as $\hat{H}_{t+1}$ is PD, it follows that final term in eq. (42) is positive, thus $K$ is PD, showing that $J^T z_{t+1} < 0$.

For the second case in which $z_t = 0$ or $\Delta_{z_{t+1}} = 0$, the system reduces to a trivial convex second order equation in $\rho$ or $\beta$ with a negative solution.

Finally, consider the case when $z_t$ and $\Delta_{z_{t+1}}$ are colinear but both non-negative. Writing $\Delta_{z_{t+1}} = \alpha z_t$ for $\alpha \in \mathbb{R}$, we note that at the minimum we have

$$
J^T z_{t+1} + \frac{1}{2} z_{t+1}^T \hat{H}_{t+1} z_{t+1} = -\frac{1}{2} \left[ \begin{array}{c} J^T \Delta_{z_{t+1}} \\ J^T z_t \end{array} \right]^T \left[ \begin{array}{c} -\beta \\ \rho \end{array} \right] = -\frac{1}{2} (\rho - \alpha\beta) J^T z_t. \tag{46}
$$

Thus at the minimum (46) is negative, closing the proof. $\qquad\square$

**Remark.** *It follows from the definition of $\rho$ and $\beta$ that if $J(w_t) = 0$, then $z_{t+1} = 0$.*

**Remark.** *If $z_{t+1} = 0$, then $z_{t+2} = -\beta J(w_{t+1})$, i.e. we reset the momentum variable $z$. This guarantees that the algorithm takes a strictly descending direction at least every two steps.*

## B  ADDITIONAL RESULTS AND IMPLEMENTATION DETAILS

### B.1  CONFIGURATIONS FOR SMALL-SCALE DATASET EXPERIMENTS

Here we provide additional details of the small-scale datasets described in sec. 4. As noted in the main paper, to give every method the best chance of working effectively we first perform a grid-search over its hyperparameters. This search is performed for each of the small-scale dataset experiments. For each first order solver listed in Table. 1, we select the configuration which achieves the lowest average error across the final ten iterations of a trajectory. The values included in the search were:

- SGD with momentum: learning rates: $\Gamma$, momentum values: $0.9, 0.95, 0.99$
- Adam: learning rates $\Gamma$, $\beta_1 : 0.9, 0.99$, $\beta_2 : 0.99, 0.999$

where $\Gamma = 0.1, 0.05, 0.01, 0.05, 0.001, 0.005, 0.0001, 0.0005$.

### B.2  HYPER-PARAMETER AND GRADIENT EVOLUTION

### B.3  RANDOM ARCHITECTURE EXPERIMENT SETUP

Each optimiser is tested on 50 random networks, that are held fixed across all methods. The number of convolutional layers is uniformly sampled between 3 and 10, and the number of channels in each layer is drawn uniformly, in powers of two, between 32 and 256. The kernel size is $3 \times 3$. Following each convolution (except the last one) there is a ReLU activation and batch-normalisation, and $3 \times 3$ max-pooling (stride 2) is placed with 50% chance. Training and evaluation is performed on CIFAR10, with a batch size of 256.

### B.4  WALL-CLOCK TIME RESULTS WITH CONJUGATE GRADIENT

### B.5  EXPERIMENTS WITHOUT A MOMENTUM HYPER-PARAMETER (FIXED $\rho = 1$)

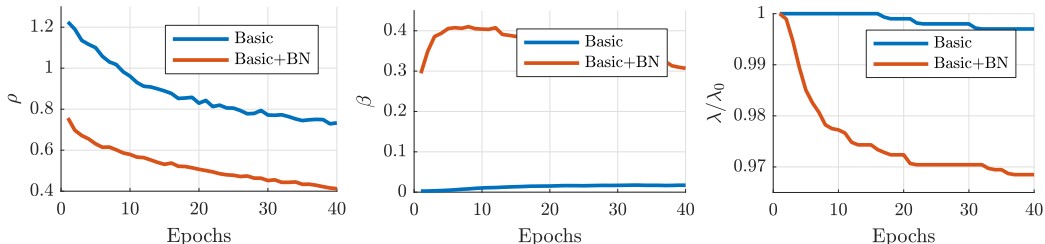

Figure 5: **Hyper-parameter evolution during training**. Average momentum $\rho$ (left), learning rate $\beta$ (middle), and trust region $\lambda$ (right), for each epoch for the basic CNN on CIFAR10, with and without batch normalisation (BN). To make their scales comparable, we plot $\lambda$ divided by its initial value (which is $\lambda_0 = 1$ with batch normalisation and $\lambda_0 = 10$ without).

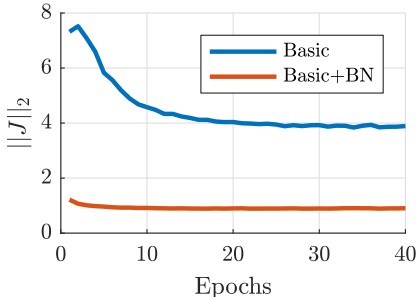

Figure 6: **Gradient evolution during training**. Average gradient norm during each epoch for the basic CNN on CIFAR-10, with and without batch normalisation (BN).

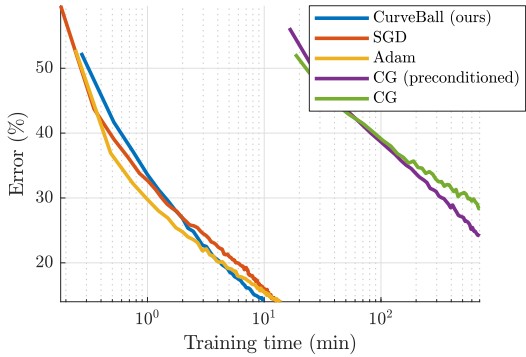

Figure 7: **Training error vs. wall clock time** (basic CIFAR-10 model). The time axis is logarithmic to show a comparison with conjugate-gradient-based Hessian-free optimisation. Due to the CG iterations, it takes an order of magnitude more time to converge than first-order solvers and our proposed second-order solver, despite the efficient GPU implementation.

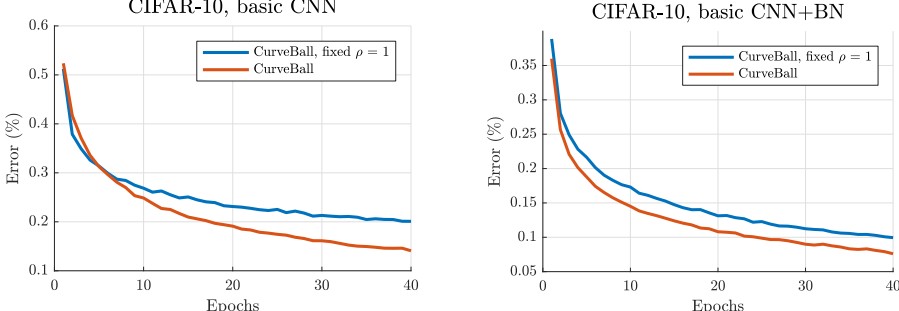

Figure 8: **Training with fixed** $\rho = 1$. Basic CNN architecture on CIFAR-10 without and with batch normalisation, respectively. Both settings use automatic tuning of the remaining hyper-parameters (by adapting eq. 18).

