# OpenReview forum: "Small steps and giant leaps: Minimal Newton solvers for Deep Learning"
_ICLR.cc/2019/Conference_

### Official Review · AnonReviewer1 · 2018-10-28
**Interesting research direction but the paper needs a lot more work before publication**

**Rating:** 3
**Confidence:** 5

**Review:**

This paper proposes an approximate second-order method with low computational cost. A common pitfall of second-order methods is the computation (and perhaps inversion) of the Hessian matrix. While this can be avoided by instead relying on Hessian-vector products as done in CG, it typically still requires several iterations. Instead, the authors suggest a simpler approach that relies on one single gradient step and a warm start strategy. The authors points out that the resulting algorithm resembles a momentum method. They also provide some simple convergence proofs on quadratics and benchmark their method to train deep neural networks.

While I find the research direction interesting, the execution is rather clumsy and many details are not sufficiently motivated. Finally, there is a lot of relevant work in the optimization community that is not discussed in this paper, see detailed comments and references below.

1) Method
The derivation of the method is very much driven on a set of heuristics without theoretical guarantees. In order to derive the update of the proposed method, the authors rely on three heuristics:
a) The first is to reuse the previous search direction z as a warm-start. The authors argue that this might be beneficial if If z does not change abruptly. In the early phase, the gradient norm is likely to be large and thus z will change significantly. One might also encounter regions of high curvature where the direction of z might change quickly from one iteration to the next.
The "warm start" at s_{t-1} is also what yields the momentum term, what interpretation can you give to this choice?

b) The second step interleaves the updates of z and w instead of first finding the optimum z. This amounts to just running one iteration of CG but it is rather unclear why one iteration is an appropriate number. It seems one could instead some adaptive strategy where CG with a fixed accuracy. One could potentially see if allowing larger errors at the beginning of the optimization process might still allow for the method to converge. This is for instance commonly done with the batch-size of first-order method. Gradually increasing the batch-size and therefore reducing the error as one gets close to the optimum can still yield to a converging algorithm, see e.g.
Friedlander, M. P., & Schmidt, M. (2012). Hybrid deterministic-stochastic methods for data fitting. SIAM Journal on Scientific Computing, 34(3), A1380-A1405.

c) The third step consists in replacing CG with gradient descent.
"If CG takes N steps on average, then Algorithm 2 will be slower than SGD by a factor of at least N, which can easily be an order of magnitude".
First, the number of outer iterations may be a lot less for the Hessian-free method than for SGD so this does not seem to be a valid argument. Please comment.
Second, I would like to see a discussion of the convergence rate of solving (12) inexactly with krylov subspace methods. Note that Lanczos yields an accelerated rate while GD does not. So the motivation for switching to GD should be made clearer.

d) The fourth step introduces a factor rho that decays z at each step. I’m not really sure this makes sense even heuristically. The full update of the algorithm developed by the author is:
w_{t+1} = w_t - beta nabla f + (rho I - beta H) (w_t - w_{t-1}).
The momentum term therefore gets weighted by (rho I - beta H). What is the meaning of this term? The -beta H term weights the momentum according to the curvature of the objective function. Given the lack of theoretical support for this idea, I would at least expect a practical reason back up by some empirical evidence that this is a sensible thing to do.
This is especially important given that you claim to decay rho therefore giving more importance to the curvature term.
Finally, why would this be better than simply using CG on a trust-region model? (Recall that Lanczos yields an accelerated linear rate while GD does not).

2) Convergence analysis
a) The analysis is only performed on a quadratic while the author clearly target non-convex functions, this should be made clear in the main text. Also see references below (comment #3) regarding a possible extension to non-convex functions.
b) The authors should check the range of allowed values for alpha and beta. It appears the rate would scale with the square root of the condition number, please confirm, this is an important detail. I also think that the constant is not as good as Heavy-ball on a quadratic (see e.g. http://pages.cs.wisc.edu/~brecht/cs726docs/HeavyBallLinear.pdf), please comment.
c) Sub-sampling of the Hessian and gradients is not discussed at all (but used in the experiments). Please add a discussion and consider extending the proof (again, see references given below).

3) Convergence Heavy-ball
The authors emphasize the similarity of their approach to Heavy-ball. They cite the results of Loizou & Richtarik 2017. Note that they are earlier results for quadratic functions such as
Lessard, L., Recht, B., & Packard, A. (2016). Analysis and design of optimization algorithms via integral quadratic constraints. SIAM Journal on Optimization, 26(1), 57-95.
Flammarion, N., & Bach, F. (2015, June). From averaging to acceleration, there is only a step-size. In Conference on Learning Theory (pp. 658-695).
The novelty of the bounds derived in Loizou & Richtarik 2017 is that they apply in stochastic settings.
Finally, there are results for non-convex functions such convergence to a stationary point, see
Zavriev, S. K., & Kostyuk, F. V. (1993). Heavy-ball method in nonconvex optimization problems. Computational Mathematics and Modeling, 4(4), 336-341.
Also on page 2, "Momentum GD ... can be shown to have faster convergence than GD". It should be mentioned that this only hold for (strongly) convex functions!

4) Experiments
a) Consider showing the gradient norms.
b) it looks like the methods have not yet converged in Fig 2 and 3.
c) Second order benchmark:
It would be nice to compare to a method that does not use the GN matrix but the true or subsampled Hessian (like Trust Region/Cubic Regularization) methods given below.
Why is BFGS in Rosenbrock but not in NN plots?
d) "Batch normalization (which is known to improve optimization)"
This statement requires a reference such as
Towards a Theoretical Understanding of Batch Normalization
Kohler et al… - arXiv preprint arXiv:1805.10694, 2018

5) Related Work
The related work should include Cubic Regularization and Trust Region methods since they are among the most prominent second order algorithms. Consider citing Conn et al. 2000 Trust Region,  Nesterov 2006 Cubic regularization, Cartis et al. 2011 ARC.
Regarding sub-sampling: Kohler&Lucchi 2017: Stochastic Cubic Regularization for non-convex optimization and Xu et al.: Newton-type methods for non-convex optimization under inexact hessian information.

6) More comments

Page 2
Polyak 1964 should be cited  where momentum is discussed.
"Perhaps the simplest algorithm to optimize Eq. 1 is Gradient Descent". This is technically not correct since GD is not a global optimization algorithm. Maybe mention that you try to find a stationary point
rho (Eq. 2) and lambda (Eq. 4) are not defined

Page 4:
Algorithm 1 and 2 and related equations in the main text: it should be H_hat instead of H.

Background
“Momemtum GD exhibits somewhat better resistance to poor scaling of the objective function”
To be precise the improvement is quadratic for convex functions. Note that Goh might not be the best reference to cite as the article focuses on quadratic function. Consider citing the lecture notes from Nesterov.

Section 2.2
This section is perhaps a bit confusing at first as the authors discuss the general case of a multivalue loss function. Consider moving your last comment to the beginning of the section.

Section 2.3
As a side remark, the work of Dauphin does not rely on the Gauss-Newton approximation but a different PSD matrix, this is probably worth mentioning.

Minor comment: The title is rather bold and not necessarily precise since the stepsize of curveball is not particularly small e.g. in Fig 1.

---

> ### Author Response · Authors · 2018-11-28
> **Short summary of response to AR1**
>
> We thank the reviewer for the constructive comments, especially with regards to having a more complete bibliography, which we have integrated.  However, we differ considerably with the reviewer in their assessment of our contribution.
>
> We would like to emphasize that our specific goal and motivation for this work was the development of a practical optimization method for large-scale deep learning.
>
> In this respect, our contribution pushes the boundaries of what has been done previously in similar papers, with a much greater scope and stringent protocol (e.g. no tuning of hyper-parameters on each experiment). We believe we are the first to apply a second-order method in such a way to extremely large settings, such as the VGG-f on ImageNet (over 60 million parameters and 1 million samples), as well as several other datasets, models, and large numbers of randomly-generated architectures.
>
> This is not to say that we do not place great value in theoretical guarantees, and in fact we proved convergence of our algorithm in convex quadratic functions (Theorem A.1) and guaranteed descent in general non-convex functions (Theorem A.2), a much broader result that the reviewer did not mention.  However, we consider that providing convergence proofs for the non-convex stochastic case (as suggested) is an unreasonable burden, both due to their much greater complexity, and because the guarantees they afford are usually mild. Instead, we only proved formally that our algorithm “does the right thing” (i.e. descends for reasonable functions), and the hard case (stochastic non-convex functions with millions of variables) is instead validated empirically.
>
> We recognise that our work places greater reliance on careful empirical evidence than the theoretical analysis preferred by the reviewer, but we hope that they will nevertheless reconsider their assessment that it represents a useful contribution to the community targeted by this conference.
>
> We will give a more detailed answer to each point in a separate comment.

---

> > ### Comment · AnonReviewer1 · 2018-11-28
> > **Authors emphasize the practical aspect of the paper but do not provide results for competing methods such as LBFGS and KFAC. The paper still lacks from a theoretical point of view.**
> >
> > First, I would like to thank the authors for providing further explanations trying to clarify their view on the empirical validity of their approach. Given their answer, it seems we would all agree that the paper is lacking from a theoretical point of view.
> >
> > Since the authors seem to stress the empirical aspect of their paper, I would find it **critical** to answer my question “Why is BFGS in Rosenbrock but not in NN plots?”. I have the same question regarding K-FAC, why is it only showed for MNIST? As a minor comment, note that KFAC seem to be reaching a lower function value in Figure 2, please use a log scale in order to improve readability.
> >
> > I disagree with your claim “We believe we are the first to apply a second-order method in such a way to extremely large settings”. BFGS has been used for a long time to optimize deep neural networks, see e.g. https://arxiv.org/pdf/1311.2115.pdf that had experiments on a a twelve layer neural network.
> >
> > Regarding the proof of convergence of Theorem A.2, note that you either require convexity or - as you suggested -, you could rely on a trust-region approach but then the decrease is only valid for achieving a **model** decrease. You would need to implement a proper trust-region algorithm to guarantee a **function** decrease. I therefore do not think the statement in Theorem A.2 is especially relevant to the deep learning setting which you seem to be targeting in this paper.

---

> > > ### Author Response · Authors · 2018-11-30
> > > **Such comparisons are not feasible at this scale; the mentioned "theory" problem is actually an implementation detail**
> > >
> > > > “It seems we would all agree that the paper is lacking from a theoretical point of view.”
> > >
> > > This is not a fair characterization of our viewpoint. Our point was that we focused much more on delivering an algorithm that is easy to implement and use by practitioners, than on tuning it to obtain theoretical guarantees (e.g. by adding variance reduction techniques). The reviewer has stated a preference for the opposite approach, which we acknowledge, and it was with this concern in mind that we included Theorems A.1-A.2 in our initial submission.
> > >
> > > > “Why is BFGS in Rosenbrock but not in NN plots?”
> > >
> > > BFGS was not included because it does not work in these settings, a widely known fact among practitioners, which is matched by our observations, and which explains its absence from the state-of-the-art in deep learning. Concretely, the issues are:
> > > - Memory. For typical problems, we simply cannot form a millions-by-millions-sized matrix. Limited-memory variants will instead require K columns, but K is still in the dozens or more. For large models, we typically cannot afford more than a factor of 2 or 3 more storage than the original parameters.
> > > - Stochasticity. BFGS breaks down with noisy functions (observed in our Stochastic Rosenbrock experiments). Variants that are robust to noise exist, but they have similar or worse memory and computation requirements.
> > >
> > > These are not a concern with stochastic first-order methods, which are widely used.
> > >
> > > > “Same question regarding K-FAC, why is it only showed for MNIST?”
> > >
> > > K-FAC requires non-trivial amounts of hand-tuning to work, as stated by the author on the project page. For example, they state the learning rate can take values between 10^-5 and 100, and in a later paper (Ba et al., ICLR 2017) they use an exponential learning rate decay with constants c_0, zeta, and exponential averaging of parameters over time.
> > >
> > > Additionally, its memory requirements are unusually large, necessitating distributed learning across multiple machines in the mentioned paper, which makes it unwieldy.
> > >
> > > Nevertheless, we show one comparison to K-FAC that is feasible, using the original paper’s code and problem setting (MLP autoencoder on MNIST), with hyper-parameters calibrated by the authors, in the interest of fairness.
> > >
> > > > “KFAC seem to be reaching a lower function value in Figure 2, please use a log scale”
> > >
> > > We will consider this scale in the final version. Note that at these noise levels it will be hard to observe any meaningful difference between the algorithms.
> > >
> > > > “I disagree with your claim “We believe we are the first to apply a second-order method in such a way to extremely large settings”. BFGS has been used for a long time to optimize deep neural networks, see e.g. https://arxiv.org/pdf/1311.2115.pdf that had experiments on a a twelve layer neural network.”
> > >
> > > We must clarify that extremely large settings mean several layers and millions of parameters, on non-toy problems. There are several aspects that make the referenced paper non-comparable:
> > > - The number of parameters or model architecture is not reported, other than mentioning that it has 12 layers.
> > > - It is an MLP applied to 28x28 inputs, which is only applicable to the least realistic scenarios.
> > > - The dataset (“CURVES”) is entirely composed of synthetic toy data.
> > >
> > > We would like to contrast this scale to training a VGG-f model with over 60 million parameters on ImageNet, with 224x224 images, among our other experiments. Hence our claim.
> > >
> > > > “Regarding the proof of convergence of Theorem A.2, note that you either require convexity or - as you suggested -, you could rely on a trust-region approach but then the decrease is only valid for achieving a **model** decrease. You would need to implement a proper trust-region algorithm to guarantee a **function** decrease.”
> > >
> > > As the reviewer has noted, this is not a problem with the proof -- which assumes a trust region, a reasonable assumption -- but with the implementation, which uses a simple mechanism for this trust region.
> > >
> > > As explained in section 3 (subsection on hyper-parameter lambda), we chose the simplest trust-region adaptation because it requires only 1 additional function evaluation. We could have easily chosen another mechanism, at the cost of speed, which is important in the large-scale. We remark that the same choice was made by Martens & Grosse (2015). This choice is entirely divorced from the validity of the proposed method, and represents one of the usual trade-offs made in any practical implementation.
> > >
> > > > “I therefore do not think the statement in Theorem A.2 is especially relevant to the deep learning setting which you seem to be targeting in this paper.“
> > >
> > > On the contrary, the extensive experiments show that the trust-region model with automatic hyper-parameter adaptation is quite accurate, otherwise the reported high performance would not have been observed.

---

> > > > ### Comment · AnonReviewer1 · 2018-12-02
> > > > **Questions in first review regarding derivation of your update + experimental results**
> > > >
> > > > Thank you for the clarifications.
> > > >
> > > > Regarding BGFS, you would of course use the limited memory variant. I do agree with what you said regarding stochasticity, second-order methods do indeed need to be stabilize for them to work. I think this does justify not showing results for this approach. On the other hand, KFAC has been used in various papers to train neural nets (see references below) so I still think the authors should provide a comparison on the larger networks.
> > > > https://arxiv.org/abs/1602.01407
> > > > https://jimmylba.github.io/papers/nsync.pdf
> > > >
> > > > Regarding the hyper-parameter lambda, this seem to make the approach rather fragile as a fixed constant lambda is not able to adapt to the curvature of the objective. Dauphin et al., 2014 also used a similar approach and although I think one should use an adaptive approach (which is actually not much more expensive),  I concede that this simple approach might work in practice after some parameter tuning, although I believe an adaptive method would be more suitable for non-convex functions.
> > > >
> > > > Now, I still have some unanswered questions regarding the derivation of your approach (questions #1 in my first review), can you please address my concerns there?
> > > >
> > > > Since we have turned our discussion to the empirical aspect of the paper, I have many more questions regarding this aspect:
> > > > 1. I would like to know if the results in Figure 2 are averaged over several runs? What does the variance look like?
> > > > 2. Can you show the plots in Figure 2 in terms of training time? In Fig. 7, your method does not outperform others so I would like to see more empirical results.
> > > > 3. How sensitive is your approach to the batch size?

---

> > > > > ### Author Response · Authors · 2018-12-05
> > > > > **Response to additional questions**
> > > > >
> > > > > Thank you for the response to our points. We have answered the initial comments that you mentioned in the appropriate thread.
> > > > >
> > > > > > “KFAC has been used in various papers to train neural nets (see references below) so I still think the authors should provide a comparison on the larger networks.”
> > > > >
> > > > > As much as we would have liked to include the improved KFAC modifications published by Grosse & Martens recently, it would represent a large departure from our training regime, which uses single GPUs. The mentioned works are all intended for multi-GPU settings. Multi-GPU synchronization of updates brings a host of confounding factors and different dynamics compared to synchronous (single-GPU) training, see Mitliagkas et al. (arXiv:1605.09774) for one example. However, we recognize that this line of research is very relevant, and we will include a caveat in the paragraph on KFAC that these improved versions exist, and should perform better in these settings.
> > > > >
> > > > > > “Results in Figure 2 are averaged over several runs? What does the variance look like?”
> > > > >
> > > > > The results in fig. 2 are not averaged over several runs, although the results in fig. 3 (left) are (averaged over 50 random architectures), which gives an indication of the variance. In practice, the variance across optimizers on larger benchmarks is very small -- we will add additional runs to the final version to make this more concrete.
> > > > >
> > > > > > “Can you show the plots in Figure 2 in terms of training time? In Fig. 7, your method does not outperform others so I would like to see more empirical results.”
> > > > >
> > > > > Yes, we can include all the plots for completeness, but the conclusions are unchanged compared to the referenced plot (which is fig. 3-right in the updated paper). Our goal was to show that it is comparable to first-order methods, despite the overhead of the second-order operations, and gains the benefit of having no hyper-parameter tuning.
> > > > >
> > > > > We would also like to bring attention to the fact that the large gap between Adam and SGD on the per-iteration plots (fig. 2) is mostly erased in the wall-clock time plot (fig. 3-right), due to the additional matrix operations that each step of Adam requires. The same phenomena happens with our method, which could be improved with better engineering of the FMAD operations.
> > > > >
> > > > > > “How sensitive is your approach to the batch size?”
> > > > >
> > > > > It is not very sensitive to batch size, as we simply used the same batch sizes as in the original papers for all of the tested architectures. It is possible that tuning the batch size brings additional benefit but we did not exploit it.

---

> ### Author Response · Authors · 2018-12-04
> **Detailed response to AR1 (part 1/2)**
>
> > “The derivation ... is very much driven on a set of heuristics without theoretical guarantees”
>
> It is common (in fact, necessary) to propose changes to existing methods based on empirical observation of their failures. It is only after proposing changes that we can prove theoretical guarantees. We would like to refer the reviewer to Theorems A.1-A.2 for such guarantees.
>
> > “In the early phase, the gradient norm is likely to be large and thus z will change significantly. One might also encounter regions of high curvature”
>
> Although the analysis (theoretical and experimental) shows that the method converges anyway, we can analyze these cases in the following way.
>
> The step z update can be rearranged as: z <- (rho*I - beta*H)*z - beta*J. Assume that the hyper-parameters and Hessian model are correct to enable convergence (H is positive-definite, rho is close to 1 and beta < 1/||H||). Then, in the high-curvature directions of H, z will be reduced the most towards 0, compared to lower-curvature directions. So in those directions, the algorithm behaves like GD. Likewise, for a large enough gradient J and low curvature, its magnitude overwhelms the first term, so the update devolves to standard GD.
>
> > “The "warm start" at s_{t-1} is also what yields the momentum term, what interpretation can you give to this choice?”
>
> The warm-starting is what makes the algorithm directly comparable to momentum. We establish this connection in section 3, and expand on differences and similarities (CurveBall vs momentum SGD). If there is a specific unclear aspect we’ll be happy to address it.
>
> > “It is rather unclear why one iteration is an appropriate number”
>
> It is appropriate due to the interleaving of steps (Algorithm 1) -- it does not represent a single isolated step, but rather builds on the previous iteration. Likewise, one could ask: why does momentum SGD only update the step z once for each iteration of w?
>
> > “Adaptive strategy where CG with a fixed accuracy”
>
> This has been done in previous works, and is very costly since it often requires dozens of steps (see fig. 7 in the appendix for a direct comparison).
>
> > “Gradually increasing the batch-size”
>
> This would introduce a schedule of batch sizes to tune, which would increase the complexity of implementation and usage.
>
> > “The number of outer iterations may be a lot less for the Hessian-free method than for SGD”
>
> The Hessian-free method would have to converge faster than SGD by orders of magnitude (measured in outer iterations) to compensate. While this is observed with linear models (where such methods are widely deployed), it is not necessarily so for deep networks, as verified in our experiments with CG (figure 7 in appendix). Nevertheless, we agree that this claim is overly broad, and we toned it down in the text.
>
> > Choice of GD over Krylov subspace methods, Lanczos
>
> There are several reasons:
> - Memory. While Krylov subspace methods have better convergence rates, they require more storage.
> - Simplicity. The aim of this paper is to create a “minimal” solver. Gradient descent fits this criteria better than the other methods; it can be described in a single line given a gradient.
> - Robustness to noise between iterations. SGD is well-understood and works well with perturbed updates; it remains to be demonstrated whether Lanczos and other methods can be made equally robust. Very recent work in this front (De Sa et al., “Accelerated Stochastic Power Iteration”, arXiv 2017) shows that much larger batches than what is acceptable for deep networks (i.e. tens of thousands) may be needed.
>
> d) “I’m not really sure [rho] makes sense”
>
> The rho parameter allows bridging two formalisms which would not be possible otherwise. Our method can be interpreted as:
> 1) A momentum GD variant: it modifies momentum GD by introducing a single term, -beta*H.
> 2) A Hessian-free optimizer variant: by performing the changes that the reviewer just mentioned (section 3).
> Note that the rho parameter is crucial for the first interpretation. Nonetheless, its apparent arbitrariness when viewed under the second interpretation can be resolved by setting rho=1. We tried fixing rho=1 experimentally, but it degrades performance; we added this experiment to the paper (fig. 8).
> The effect of rho can be interpreted in two ways. First, rho<1 gradually erases stale updates (based on old Hessian matrices) from the z buffer, which is important for a non-quadratic objective. Second, it results in the regularizer (1-rho)*||z||^2 in the quadratic model, which can be beneficial, and has small magnitude with rho close to 1.
> Finally, the automatic hyper-parameter tuning (eq. 18) requires rho to be present, which is another practical reason for its presence.

---

> ### Author Response · Authors · 2018-12-04
> **Detailed response to AR1 (part 2/2)**
>
> > “You claim to decay rho”
>
> We do not claim to explicitly decay rho (one of the points we focused on was on not having complicated schedules to tune), but rather it decays naturally as a consequence of the automatic hyper-parameter adaptation (fig. 5 in updated paper).
>
> > “Better than simply using CG on a trust-region model?”
>
> We tried this variant (fig. 7), but the large number of inner iterations makes it less competitive.
>
> > “Analysis is only performed on a quadratic”
>
> Theorem A.2 deals specifically with non-quadratic functions.
>
> > “It appears the rate would scale with the square root of the condition number”; “constant is not as good as Heavy-ball on a quadratic”
>
> Unfortunately, the rates that we derived are not as directly interpretable as SGD or momentum SGD, despite our best efforts (eq. 38). However, on a convex quadratic we do not expect our rate of convergence to be better than momentum SGD with the optimal momentum parameter. Note that using momentum GD with the optimal hyper-parameters (which require knowledge of the Hessian eigenvalues of the quadratic) already provides a square-root improvement in the condition number compared to GD. We would consider any further improvement in linear convergence in this well-explored setting to be quite a breakthrough.
>
> > “Sub-sampling … not discussed”; “consider extending the proof”
>
> We addressed this matter in our initial response.
>
> > “Consider showing the gradient norms”
>
> We added this to the paper (fig. 6).
>
> > “Methods have not yet converged”
>
> It is standard practice in deep learning comparisons to give methods a budget of epochs to converge over, due to the time-consuming nature of the experiments. We used the default numbers of epochs for which the SGD learning rate schedules were defined. Note that even when this is not the case, early-stopping is used as an effective regularization method (since the parameters vastly outnumber the samples).
>
> > Compare to Newton method with true Hessian
>
> We added this to the paper; see the “Exact Hessian” (Newton method) row in table 1 and fig. 1.
>
> > “Why is BFGS in Rosenbrock but not in NN plots?”
>
> This was addressed in another comment.
>
> > “Dauphin does not rely on the Gauss-Newton approximation”
>
> We cited Dauphin et al. as a reference on avoiding saddle-points with PSD surrogates for the Hessian. We did not mean to imply that they used the Gauss-Newton surrogate. We agree that this should be more clear, and corrected the text.
>
> > “The title is rather bold and not necessarily precise since the stepsize of curveball is not particularly small”
>
> The title is actually a reference to the different step sizes in different parameter-space directions, when optimizing an ill-conditioned function (mentioned in the 2nd paragraph of section 2.1).
>
> > Additional citations and other editing suggestions
> We would like to thank the reviewer for the suggestions, which we incorporated in the paper. Note that we cited Loizou & Richtarik (2017) as an up-to-date summary of theoretical results; nevertheless we now cite the original papers.

---

### Official Review · AnonReviewer2 · 2018-10-31
**Good Paper, Accept**

**Rating:** 7
**Confidence:** 4

**Review:**

In this paper, the authors introduce a new second-order algorithm for training deep networks. The method, named CurveBall, is motivated as an inexpensive alternative to Newton-CG. At its core, the method augments the update role for SGD+M with a Hessian-vector product that can be done efficiently (Algorithm 1). While a few new hyperparameters are introduced, the authors propose ways by which they can be calibrated automatically (Equation 16) and also prove convergence for quadratic functions (Theorem A.1) and guaranteed descent (Theorem A.2). The authors also present numerical results showing improved training on common benchmarks. I enjoyed reading the paper and found the motivation and results to be convincing. I especially appreciate that the authors performed experiments on ImageNet instead of just CIFAR-10, and the differentiation modes are explained well. As such, I recommend the paper for acceptance.


I suggest ways in which the paper can be further improved below:

- In essence, the closest algorithm to CurveBall is LiSSA proposed by Agarwal et al. They use a series expansion for approximating the inverse whereas your work uses one iteration of CG. If you limit LiSSA to only one expansion, the update rule that you would get would be similar to that of CurveBall (but not exactly the same). I feel that a careful comparison to LiSSA is necessary in the paper, highlighting the algorithmic and theoretical differences. I don't see the need for any additional experiments, however.
- For books, such as Nocedal & Wright, please provide page numbers for each citation since the information quoted is across hundreds of pages.
- It's a bit non-standard to see vectors being denoted by capital letters, e.g. J(w) \in R^p on Page 2. I think it's better you don't change it now, however, since that might introduce inadvertent typos.
- It would be good if you could expand on the details concerning the automatic determination of the hyperparameters (Equation 16). It was a bit unclear to me where those equations came from.
- Could you plot the evolution of \beta, \rho and \lambda for a couple of your experiments? I am curious whether our intuition about the values aligns with what happens in reality. In Newton-CG or Levenberg-Marquardt-esque algorithms, with standard local strong convexity assumptions, the amount of damping necessary near the solution usually falls to 0. Further, in the SGD+M paper of Sutskever et al., they talked about how it was necessary to zero out the momentum at the end. It would be fascinating if such insights (or contradictory ones) were discovered by Equation 16 and the damping mechanism automatically.
- I'm somewhat concerned about the damping for \lambda using \gamma. There has been quite a lot of work recently in the area of Stochastic Line Searches which underscores the issues involving computation with noisy estimates of function values. I wonder if the randomness inherent in the computation of f(w) can throw off your estimates enough to cause convergence issues. Can you comment on this?
- It was a bit odd to see BFGS implemented with a cubic line search. The beneficial properties of BFGS, such as superlinear convergence and self-correction, usually work out only if you're using the Armijo-Wolfe (Strong/Weak) line search. Can you re-do those experiments with this line search? It is unexpected that BFGS would take O(100) iterations to converge on a two dimensional problem.
- In the same experiment, did you also try (true) Newton's method? Maybe we some form of damping? Given that you're proposing an approximate Newton's method, it would be a good upper baseline to have this experiment.
- I enjoyed reading your experimental section on random architectures, I think it is quite illuminating.
- Please consider rephrasing some phrases in the paper such as "soon the latter" (Page 1), "which is known to improve optimisation", (Page 7), "non-deep problems" (Page 9).

---

> ### Author Response · Authors · 2018-11-15
> **Response to AR2**
>
> Thank you for the very thoughtful suggestions and questions.
>
> > Comparison to the LiSSA algorithm
> There are indeed very interesting connections between CurveBall and LiSSA. Despite their main update being derived from very different assumptions, we found that it is possible to manipulate it into a form that is directly comparable to ours, without a learning rate \beta. Another difference is structural: LiSSA uses a nested inner loop to create a Newton-like update from scratch every iteration, like other Hessian-free methods, while our algorithm structure has no such nesting and thus has the same structure as momentum SGD (cf. Alg. 1 and Alg. 2 in the paper).
>
> We updated the paper with a much more detailed exposition of these points, which we have only hinted at in this response to keep it short. It can be found in the last (large) paragraph of the related work (section 5, p. 9).
>
> > Page numbers on books
> Thank you, we agree that this is important; we just added them to the paper.
>
> > Vectors as capital letters, e.g. J(w)
> We share this concern, however this was used to simplify our exposition of automatic differentiation (sec. 2.2). There, the gradient J arises from the multiplication of several Jacobians, which are generally matrices, and it only happens to be a vector because of the shape of the initial projection. We could have treated Jacobians and gradients separately, but it would hamper this unifying view which we found more instructive.
>
> > Automatic hyper-parameters derivation
> Although this can be found in the work of Martens & Grosse (2015), to make the paper self-contained we added the derivations to the appendix (section A.1), consisting of a simple minimization problem in the \rho and \beta scalars.
>
> > “Plot the evolution of \beta, \rho and \lambda”
> This is an interesting aspect to analyze. We plot these quantities for two models (with and without batch normalization) in the (newly-added) Fig. 5.
>
> It seems that the momentum hyper-parameter \rho starts with a high value and decreases over time, with what appears to be geometric behavior. This is in line with the mentioned theory, although it is simply a result of the automatic tuning process.
>
> As for the learning rate \beta, it increases in an initial phase, only to decrease slowly afterwards. We can compare this to the practitioners’ manually-tuned learning rate schedules for SGD that include “burn-in” periods, which follow a similar shape (He et al., 2016). Similar schedules were also obtained by previous work on gradient-based hyper-parameter tuning (Maclaurin et al., “Gradient-based Hyperparameter Optimization through Reversible Learning”, ICML 2015).
>
> The trust region \lambda decreases over time, but by a minute amount. The trust region adaptation is a 1D optimization problem over \lambda, minimizing the difference between the ratio \gamma and 1. This 1D problem has many local minima, punctuated by singularities corresponding to Hessian eigenvalues (see Wright & Nocedal (1999) fig. 4.5). Given a large enough spread of eigenvalues, it is not surprising that a minimum close to the initial \lambda was found by the iterative adaptation scheme.
>
> > Comment on Stochastic Line Searches; damping for \lambda using \gamma
> We agree that a more satisfactory solution would be to employ the ideas of Probabilistic Line Search. However, it would involve reframing the optimization in Bayesian terms, which would be a large change and add significant complexity, which we tried to avoid.
>
> Instead, and inspired by KFAC (Martens & Grosse, 2015), we change \lambda in *small* increments based on how close \gamma is to 1. The argument is that, even if a particular batch gives an inaccurate estimate of \gamma, in expectation it should be correct, and so most of the small \lambda increments will be in the right direction (in 1D). The procedure would indeed be unstable if the increments were much less gradual.
>
> > Re-do experiments with Armijo-Wolfe line search; BFGS performance
> BFGS only needs 19 function evaluations to achieve 10^-4 error on the *deterministic* Rosenbrock function, which we considered to be a reasonable result. However, the *stochastic* Rosenbrock functions are more difficult, as expected.
>
> The cubic line search is part of the BFGS implementation that ships with Matlab. Following this suggestion, we also tested minFunc’s implementation of L-BFGS, which includes Armijo and Wolfe line searches. We tried several initialization schemes, as well as different line search variants, and found no improvement over the previous ones (Table 1).
>
> > “Try (true) Newton's method”
> We considered LM as the upper baseline as the Hessian isn’t necessarily definite positive, but the true Newton’s method is indeed subtly different. It achieves slightly better results overall (see updated Table 1). Note that when the Hessian matrix has negative eigenvalues, we use the absolute values instead.
>
> > “Please consider rephrasing some phrases”
> We did; thank you for the suggestions.

---

### Official Review · AnonReviewer3 · 2018-11-07
**Well-motivated idea**

**Rating:** 7
**Confidence:** 5

**Review:**

Authors propose choosing direction by using a single step of gradient descent "towards Newton step" from an original estimate, and then taking this direction instead of original gradient. This direction is reused as a starting estimate for the next iteration of the algorithm. This can be efficiently implemented since it only relies on Hessian-vector products which are accessible in all major frameworks.

Based on the fact that this is an easy to implement idea, clearly described, and that it seems to benefit some tasks using standard architectures, I would recommend this paper for acceptance.

Comments:
- introducing \rho parameter and solving for optimal \rho, \beta complicates things. I'm assuming \rho was needed for practical reasons, this should be explained better in the paper. (ie, what if we leave rho at 1)
- For  ImageNet results, they show 82% accuracy after 20 epochs on full ImageNet using VGG. Is this top5 or top1 error? I'm assuming top5 since top1 would be new world record for the number of epochs needed. For top5, it seems SGD has stopped optimizing at 60% top5. Since all the current records on ImageNet are achieved with SGD (which beats Adam), this suggests that the SGD implementation is badly tuned
- I appreciate that CIFAR experiments were made using standard architectures, ie using networks with batch-norm which clearly benefits SGD

---

> ### Author Response · Authors · 2018-11-09
> **Response to AR3**
>
> We would like to thank the reviewer for the comments and questions.
>
> > “Introducing \rho parameter and solving for optimal \rho, \beta complicates things”
> It does, but it makes the proposed solver more reliable (no tuning is necessary).
>
> It is possible to set the hyper-parameters manually, but this requires multiple runs (similarly to learning rate tuning for SGD), which makes it much less convenient.
>
> One reason for introducing rho, in addition to the connection to momentum SGD, is that this allowed us to use the same automatic tuning strategy as Martens & Grosse (2015). This formulation depends on the update equations having both rho and beta hyper-parameters. Another intuitive reason is to slowly forget stale updates, which is the same role played by this parameter in momentum SGD. We will clarify this further in the paper.
>
> Our analysis for the convex quadratic case (visualized in fig. 4, appendix A) shows that the algorithm converges on a relatively large region of the (rho, beta) parameter-space. However, the best performance is achieved in a relatively narrow band, which will vary depending on the Hessian eigenvalues. Automatically solving for the optimal rho and beta removes this concern.
>
> > “For ImageNet results, they show 82% accuracy after 20 epochs on full ImageNet using VGG. Is this top5 or top1 error?”
> This is top-1 training error; if it were top-1 validation error, it would indeed be unreasonably good.
>
> Counter-intuitively, SGD is well tuned -- its training error stalls, however the validation error keeps going down for a few more epochs. The learning rate annealing schedule was chosen by the authors of the VGG-f model taking this into account. This is a problem with SGD -- as it is implemented, it works both as optimizer and regularizer.
>
> We show the training error in all plots in order to accurately measure improvements in optimization, without the added confusion of such regularization effects. We study and discuss the validation error separately (in the last subsection of the experiments).
>
> In summary, we found that models that have an appropriate number of parameters w.r.t. the dataset size benefit from our improved optimization, while the larger models (e.g. ResNet) require additional regularization to lower the validation error.
>
> We view this development as a two-step process: first we create algorithms that can optimize the objective function efficiently; and once we have them, we can focus on effective regularization techniques. We believe that this strategy is more promising than developing both simultaneously.

---

### Public Comment · ~Tim_Cooijmans1 · 2018-12-03
**What is the relationship to single-step HF and Nesterov momentum?**

How does curveball relate to the single-step-CG variant of HF discussed in the cited paper "On the importance of initialization and momentum in deep learning" (Sutskever et al 2013)? From the discussion in that paper, I wish the authors of the present paper would also include Nesterov momentum in their experiments.

---

> ### Author Response · Authors · 2018-12-06
> **Relationship to Nesterov accelerated GD**
>
> Thank you for pointing out this interesting connection. Quoting from the paper you mentioned:
>
> “If CG terminated after just 1 step, HF becomes equivalent to NAG, except that it uses a special formula based on the curvature matrix for the learning rate instead of a fixed constant.”
>
> The same reasoning applies to our own method, since it is a modification of the HF approach. The main difference from NAG is then the use of curvature, which Nesterov’s accelerated GD (NAG) does not use, as pointed out in the quote.
>
> As a practical matter, note that NAG is not robust to perturbations and accumulates errors in the gradient oracle linearly with iterations (as discussed in a paper by Nesterov himself and co-authors, “First-Order Methods of Smooth Convex Optimization with Inexact Oracle”).
>
> NAG is a part of standard deep learning frameworks and practitioners try it from time to time. While we cannot comment on its success in RNNs with sigmoids (which Sutskever focused on), our experience with modern CNNs shows that it is “hit or miss”, with no clear advantage over a similarly well-tuned momentum SGD. Its success is highly dependent on the stochasticity of the problem and many other factors (which is expected given the Nesterov paper we mentioned).

---

### Meta-Review · Area_Chair1 · 2018-12-07
**a sensible proposal, but little evidence of optimization benefit**

**Confidence:** 5
**Recommendation:** Reject

**Metareview:**

The proposal is a scheme for using implicit matrix-vector products to exploit curvature information for neural net optimization, roughly based on the adaptive learning rate and momentum tricks from Martens and Grosse (2015). The paper is well-written, and the proposed method seems like a reasonable thing to try.

I don't see any critical flaws in the methods. While there was a long discussion between R1 and the authors on many detailed points, most of the points R1 raises seem very minor, and authors' response to the conceptual points seems satisfactory.

In terms of novelty, the method is mostly a remixing of ideas that have already appeared in the neural net optimization literature. There is sufficient novelty to justify acceptance if there were strong experimental results, but in my opinion not enough for the conceptual contributions to stand on their own.

There is not much evidence of a real optimization improvement. The per-epoch improvement over SGD is fairly small, and (as the reviewers point out) probably outweighed by the factor-of-2 computational overhead, so it's likely there is no wall-clock improvement. Other details of the experimental setup seem concerning; e.g., if I understand right, the SGD training curve flatlines because the SGD parameters were tuned for validation accuracy rather than training accuracy (as is reported). The only comparison to another second-order method is to K-FAC on an MNIST MLP, even though K-FAC and other methods have been applied to much larger-scale models.

I think there's a promising idea here which could make a strong paper if the theory or experiments were further developed. But I can't recommend acceptance in its current form.

---

> ### Author Response · Authors · 2019-04-10
> **Response to final decision**
>
> We would like to address a few points which we consider inaccurate in the AC’s final decision:
>
> > “The per-epoch improvement over SGD is fairly small”
> We show improvements on the order of 5% on CIFAR, and even reduce the ResNet error by a factor of 3 (from an already low 2.1% to 0.7%). We also improve on Adam by 2 to 3% in most cases. Compare this to many papers in the same venue that were published with reductions of 1% error compared to their baselines.
>
> > “[Improvements] probably outweighed by the factor-of-2 computational overhead, so it's likely there is no wall-clock improvement”
> We show that there is a wall-clock improvement in Fig. 3. The improvements in optimization compensate for the small computational overhead.
>
> > “SGD parameters were tuned for validation accuracy”
> This was only the case for the ImageNet experiments, which are 2 out of 7 experimental settings. However, we agree that it is fairer to use the same cross-validation protocol as for the other experiments. We found that SGD with the optimal learning rate still does not outperform Adam or our method, and will include this change in a future version.
>
> > “roughly based on [...] tricks from Martens and Grosse (2015).”
> The core of our method is an implicit inversion of the Hessian, while the mentioned work has an explicit model of the Hessian that is kept in memory - the methods differ substantially.